# Evaluating Age-Related Anatomical Consistency in Synthetic Brain MRI against Real-World Alzheimer's Disease Data.

**Hadya Yassin**[1,2]                                     HADYA.YASSIN@HPI.DE
**Jana Fehr**[1,2,3]                                   JANA.FEHR@BIH-CHARITE.DE
**Wei-Cheng Lai**[1,2]                                 WEI-CHENG.LAI@HPI.DE
**Alina Krichevsky**[1,2]                          ALINA.KRICHEVSKY@STUDENT.HPI.DE
**Alexander Rakowski**[1,2]                        ALEXANDER.RAKOWSKI@HPI.DE
**Christoph Lippert**[1,2,4]                         CHRISTOPH.LIPPERT@HPI.DE

[1] *Digital Engineering Faculty, University of Potsdam, Germany*

[2] *Digital Health and Machine Learning, Hasso-Plattner-Institute, Potsdam, Germany*

[3] *QUEST Center for Responsible Research, Berlin Institute of Health (BIH), Charité Universitätsmedizin Berlin, Germany*

[4] *Hasso Plattner Institute for Digital Health at Mount Sinai, Icahn School of Medicine at Mount Sinai, New York, NY, United States of America*

**Editors:** Accepted for publication at MIDL 2024

## Abstract

This study examines the realism of medical images created with deep generative models, specifically their replication of aging and Alzheimer's disease (AD) related anatomical changes. Previous research focused on developing generative methods with limited attention to image fidelity. We aim to assess the resemblance of brain MRI generated by a StyleGAN3 model with causal controls to neurodegenerative changes. For a benchmark, we conducted a visual Turing test (VTT) to see if radiologists could distinguish between synthetic and real images. Then, we employed a U-Net-based model to segment hallmarks relevant to normal aging and AD. Finally, we conducted statistical tests for our hypothesis that no significant differences existed between real and synthetic images. VTT results showed radiologists struggled to differentiate between image types, highlighting VTT's limitations due to subjectivity and time constraints. We found slight hippocampus distribution differences ($P = 5.7$e-2) and significant lateral ventricle discrepancies ($Ps < 5.0$e-2), indicating higher hippocampus realism and ventricle size inconsistencies. The model more effectively simulated changes in the hippocampus than in the lateral ventricles, where difficulties were encountered with certain subgroups. We conclude that the VTT alone is inadequate for a comprehensive quality evaluation, promoting a more objective approach. Future research could adapt our approach to evaluate other generated medical images intended for different downstream tasks. For reproducibility, we provide detailed code implementation[1].

**Keywords:** Evaluating Generative Models, MRI, Alzheimer's, Anatomical Consistency.

## 1. Introduction

Deep learning algorithms, essential for automating medical image analysis, rely heavily on access to abundant, high-quality datasets for training (Sarker, 2021). Acquiring such data is hindered by high costs associated with equipment, expert annotation, subject availability,

---

1. github.com/hsyassin/BioPlausibleSynthImgEval.git

and privacy concerns (Diaz et al., 2021; Kaissis et al., 2020). Consequently, these challenges can lead to data biases that significantly impair the algorithms' performance in real-world applications, as they may not accurately reflect the diversity of real patient populations or the complexity of medical conditions encountered in clinical settings (Nittas et al., 2023).

Despite significant progress in synthetic medical image generation through generative adversarial networks (GANs) (Dash et al., 2023) and Diffusion models (Yang et al., 2023), traditional evaluation metrics like the Fréchet inception distance (FID) and Structural Similarity Index Measure (SSIM) primarily assess overall image quality, overlooking anatomical accuracy critical for medical applications. Thus, accurately mirroring human physiology and conditions like AD is crucial for synthetic images to bridge dataset gaps. Therefore, we need "Biological Plausibility" metrics, introduced by (Treder et al., 2022) and designed to provide anatomical analysis specific to clinical needs. Our evaluation method focuses on crucial neurodegenerative features for normal aging, which are more prominent in AD, including ventricular enlargement and hippocampal shrinkage (Frisoni et al., 2010; Katabathula et al., 2021).

VTT is a standard for assessing synthetic images' biological plausibility in the generative field, as utilized by Khader et al. (2022) to evaluate diffusion model performance. However, Treder et al. (2022) highlights VTT's limitations: subjectivity, high costs, and difficulty detecting subtle anatomical changes. These issues suggest that VTT is insufficient in ensuring synthetic images' clinical applicability. Our study benchmarks against VTT, promoting more objective, robust, and clinically relevant evaluation methods to improve synthetic images' clinical utility.

Longitudinal GAN studies on AD explore biological plausibility. Xia et al. (2021) analyzes relative changes (RC) in the region of interest (ROI) volumes between baseline and follow-up images, real and synthetic. Peng et al. (2021) examines the ROI Absolute Volume Difference (AVD) between synthetic follow-ups and real manual segmentation. Fu et al. (2023) calculate mean absolute error (MAE) for ROI volumes in longitudinal data, differing from Ravi et al. (2022) that calculate MAE for randomly matching samples by age, sex, and clinical dementia rating (CDR). The pairwise approach in longitudinal studies, which tracks changes in the same individual, offers more accuracy than random matching. Ribeiro et al. (2023) evaluates MAE using model-predicted counterfactual volumes rather than actual volumes segmented from the counterfactual images, which may misrepresent true volumes. These methods aim to compare anatomical consistencies between real and synthetic data but reduce the evaluation to a single average value (RC, AVD, MAE), favoring benchmark comparisons of generative models over anatomical consistency and clinical relevance. Wilms et al. (2022) investigates age-related brain volume changes across generative models without real data comparison, omitting key evaluations of synthetic-to-real anatomical consistency.

Additionally, the mentioned methods segment ROIs using statistical and algorithmic strategies, such as FSL (FMRIB Software Library) and FreeSurfer (Jenkinson et al., 2012; Fischl, 2012), or multi-atlas methods (Wang et al., 2014; Doshi et al., 2013). In contrast, our method utilizes the advanced capabilities of deep learning. Litjens et al. (2017); Shen et al. (2017) demonstrate that deep learning surpasses traditional methods in accuracy, efficiency, and detection of complex patterns.

Deep learning segmentation enables accurate anatomical consistency comparisons between synthetic ROI sizes and real counterparts, capturing essential variability in aging and AD. Such analysis is crucial for clinical applications requiring high accuracy and transparency. Our objective is to assess the null hypothesis (H0) that there is no significant distributional difference in brain ROI areas between real Alzheimer's disease neuroimaging initiative (ADNI) images and their synthetic counterparts. We utilize statistical tests to examine differences in magnitude, certainty, direction, and distribution (dist) shapes. Finally, Jung et al. (2023); Xia et al. (2021) conducted statistical analyses to identify dist differences and focused specifically on anatomical consistencies across CDR groups using multi-hypothesis testing. However, our research expands to include age and sex covariates and employs an alternative approach that avoids multi-hypothesis testing.

## 2. Methodology:

Biological plausibility overview in Figure 1. Input: AD-focused Synthetic images generated by a conditional GAN with causal control and a subset of the ADNI dataset as the real-world reference (unseen in training and validation of both generative and segmentation models to reduce bias). VTT was selected as a benchmark for its established role in assessing synthetic image realism. Our method quantifies ROI area via a segmentation model and performs statistical tests to evaluate anatomical consistency between image types across different covariates. Understanding our analysis requires familiarity with real-world and synthetic data characteristics, synthetic image generation, and the generative model's capabilities.

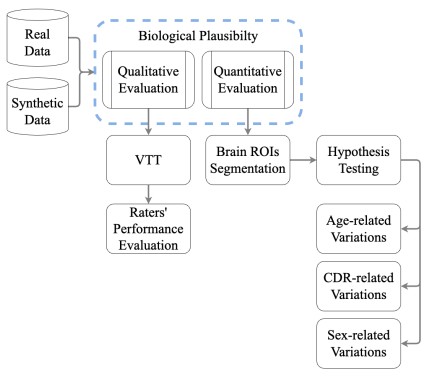

Figure 1: Overview of the analysis

### 2.1. Use Case and Datasets:

**Real Data.** We sourced real magnetic resonance imaging (MRI) brain scans from the UK Biobank (UKB) (Sudlow et al., 2015) and ADNI (Alzheimer's Disease Neuroimaging Initiative (ADNI), 2021) datasets. The UKB provided 42,427 high-resolution T1-weighted 3T MRI brain scans of individuals aged 40-69 (mean age: 55). Unlike the UKB, which serves a general cohort, the ADNI provides a focused dataset on specific diseases. The ADNI database (adni.loni.usc.edu), established in 2003, explores the use of imaging, biological markers, and clinical assessments to track the progression of mild cognitive impairment (MCI) and early AD. ADNI consists of high-resolution T1-weighted 3T MRI brain scans from 9,183 participants aged 55-90 (mean age: 74), including 3,273 cognitive normal (CN) cases (CDR=0), 4,943 MCI cases (CDR=0.5), and 967 AD cases (CDR$\geq$1).

**Synthetic Data.** We generated 600,000 2D mid-slice images using a conditional style-based generative adversarial network (StyleGAN3) model (Karras et al., 2021) with causal control (Pawlowski et al., 2020). While Kocaoglu et al. (2017) introduced causal inference to

StyleGAN3, our model combines a conditional GAN structure (Mirza and Osindero, 2014; Miyato and Koyama, 2018) with a custom causal model. To achieve a high-quality synthesis, we initially trained the model on the UKB dataset for its extensive data volume, followed by fine-tuning it on the ADNI dataset. For both datasets, our causal model consistently incorporated covariates like **age**, **sex**, and **left and right lateral ventricle volumes**. Given the disease-specific nature of the ADNI, we added AD-associated conditional labels like **CDR** and **hippocampus volumes** (Frisoni et al., 2010; Katabathula et al., 2021) during its fine-tuning.

During the inference step, the images are synthesized by the causal and StyleGAN3 models. The causal model is used to provide conditional labels as vectors, *i.e.*, **age**, **sex**, **CDR**, **lateral ventricle volumes**, **cerebral cortex volumes**, and **hippocampus volumes**. Taking advantage of the causal model allows us to control labels by changing ages, resulting in different volumes since our assumption is that age causally affects brain volumes. These conditional labels are fitted into the StyleGAN3 model to synthesize images with specific attributes related to age, sex, and either healthy or AD diagnoses, as indicated by CDR values, for more details on the conditional causal model (Appendix A).

### 2.2. Benchmark: Visual Turing Test (VTT)

To assess synthetic image realism, we conducted a VTT with two radiologists reviewing 100 image pairs: 50 from UKB and 50 from ADNI with their synthetic counterparts. Each pair had one real and one synthetic image, matched by **age**, **sex**, and **CDR** for ADNI images only. The radiologists identified the synthetic image, rated their confidence, and explained their choices, enabling a comprehensive realism evaluation. The test, done with ImFusion Labels software (ImF), was untimed to allow breaks and resumptions.

### 2.3. Biological Plausibility Analysis

**ROI Quantification via Segmentation.** Our generative model produces 2D mid-slice brain MRI, while established deep-learning segmentation models like SynthSeg (Billot et al., 2023) are designed for 3D volumes. To bridge this gap, we developed a 2.5D segmentation model suitable for 2D and 3D data. Using SynthSeg, we generated ground truth masks for UKB and ADNI datasets. To prevent bias in our segmentation towards ADNI (real-world reference), we trained and validated state-of-the-art (SOTA) architectures solely on UKB.

During inference, the best-performing model (adapted UNet (A-UNet)) was selected for its highest median dice similarity coefficient (DSC) from external ADNI testing. This model, an adaptation of the original UNet (Ronneberger et al., 2015), was optimized by increasing initial features from 32 to 64 and adding a layer of depth. Details on model implementation and comparisons with other architectures are in Appendix C. We quantified ROI areas by counting pixels in segmented masks, then normalized these areas against the intracranial area to account for brain size variations. Min-max scaling was applied to these normalized values to ensure consistency with observed area ranges.

**Statistical Analysis.** In the VTT, a **two-tailed binomial test** evaluated radiologists' accuracy against random guessing and **Cohen's kappa** for inter-rater agreement. To evaluate anatomical consistency between image types, we used statistical tests to explore the

hypothesis that no significant differences exist in the ROI dists. between real and synthetic images, setting the significance level at 5.0e-2. The **permutation test (permute)**, chosen for its non-parametric nature that requires no dist assumptions, assesses the statistical significance of mean differences through repeated recalculations, minimizing false significance risks. Subsequently, **Cohen's d** was used to measure **effect size (ES)** and direction. Additionally, we employed a 95% **confidence interval (CI)**, based on recommendations from Lee (2016), to assess ES estimation precision and indicate the statistical significance of our findings. Finally, The **Kolmogorov-Smirnov test (KS)** further compared dist shapes, providing added insights into discrepancies between real and synthetic image dists.

## 3. Experiments and Results

**Experiment Setup.** Our experiments evaluated the impact of aligning covariate dists. (age, CDR, and sex) on image analysis, maintaining a 5:1 synthetic-to-real ROI area ratio throughout the study. We conducted experiments on matched and mismatched covariate dists, where we would expect a higher deviation between real and synthetic ROI dists. when deviating from matching dists. The experiments are: a) matched covariate dists, b) introduced variances by adjusting the median age of the synthetic samples' dist by $\pm5$ years, c) CDR, and d) sex mismatches in synthetic dist are detailed in Appendix D.

**VTT Results.** Radiologist 1 (R1) achieved a 51% accuracy rate, similar to random guessing ($P =$ 9.2e-1), while radiologist 2 (R2) scored 15%, a significant difference ($P =$ **4.8e-13**). Cohen's kappa of 0.167 indicates low agreement between the two, suggesting variability in image perception and image fidelity conclusions. R1 often indicated "moderate certainty" and cited "contrast problems," while R2 often reported "very low certainty" and highlighted "anatomical inaccuracies" as their certainty level and reasons behind most of their decisions. Interestingly, neither of them considered an "inaccurate representation of pathology." A follow-up revealed R2's bias stemming from MRI characteristics familiarity, leading to misclassifications by associating smoothing effects with real images despite their presence in both image types. Figure 3 a) and Figure 6 in Appendix B display a misclassification by both R2 and R1 for the same pair of images, but for different reasons. R2 stated "anatomical inaccuracies", while R1 stated "noise patterns", despite the image being real. The sole instance where both radiologists strongly agreed and accurately identified a synthetic image, citing inaccurate anatomy due to unusually small ventricles, is shown in Figure 3 b). Further VTT results are in Appendix B.

**Biological Plausibility Results.** Regression analysis in Figure 2 reveals that Lateral Ventricles (LV) areas increase and Hippocampus (HC) areas decrease with age, aligning with neurodegeneration patterns in both real and synthetic images. Furthermore, Table 1 demonstrates significant differences for LV in **matched dists.** ($P =$ **2.0e-3**, **negative** $CI -_l -_u$), indicating synthetic images overestimate LV's enlargement with age compared to real images (Negative ES, see Figure 2). The LV KS test results show significant shape differences in dists. across **both matched and unmatched conditions** (Figure 8). In contrast, hippocampus results show no significant difference in permute tests. However, a near-significant $p$ (5.7e-2) and positive ES (7.0e-2) suggest minor overestimation in synthetic image ROI size reduction with age at matched dists. Analysis in Figure 2 shows minimal

differences between image types in LV for the 75-80 and 80-85 age groups, with nearly identical box plots. For HC, the 60-65 and 85-90 age groups have lower medians of synthetic areas, while the 80-85 group has a higher median.

Table 1: Statistical Comparison between Real vs. Synthetic ROI Areas Across Distributions. **Match:** same covariate dist for Image Types (Median age of 74); **Mismatch:** Synthetic dist age median shifts by $\pm$ **3 or 5 years**). $CI -_l -_u$: CI bounds. **Significant differences** are highlighted.

| ROI | Cond. | Permute | ES | $CI_l$ | $CI_u$ | KS |
|-----|-------|---------|-----|--------|--------|-----|
| LV | -5 | 6.0e-2 | 7.1e-2 | -5.8e-3 | 1.5e-1 | **1.3e-2** |
| LV | -3 | 8.2e-1 | 9.5e-3 | -6.7e-2 | 8.6e-2 | **1.4e-2** |
| LV | Match | **2.0e-3** | -1.3e-1 | **-2.0e-1** | **-5.0e-2** | **2.0e-8** |
| LV | +3 | **0.0e-0** | -2.3e-1 | **-3.1e-1** | **-1.6e-1** | **2.1e-14** |
| LV | +5 | **5.0e-2** | -3.1e-1 | **-3.9e-1** | **-2.3e-1** | **4.4e-16** |
| HC | -5 | **4.0e-3** | -1.1e-1 | **-1.9e-1** | **-3.6e-2** | **1.1e-2** |
| HC | -3 | 3.2e-1 | -4.0e-2 | -1.2e-01 | 3.7e-2 | 4.5e-1 |
| HC | Match | 5.7e-2 | 7.0e-2 | -6.7e-3 | 1.5e-1 | 8.7e-2 |
| HC | +3 | **4.0e-3** | 1.1e-1 | **3.8e-2** | **1.9e-1** | **2.4e-3** |
| HC | +5 | **5.0e-2** | 2.1e-1 | **1.4e-1** | **2.9e-1** | **9.4e-8** |

**Mismatched Conditions** with age adjustments (+3, +5) showed significant differences in LV size, with larger negative ES than the matched dist. Conversely, deviations of (-3, -5) lacked significant findings and had smaller ES, particularly at -3, confirming the generative model's overestimation of LV sizes in matched dists. (As seen in Figure 2). For HC, the -3 mismatch, like matched dist, showed no significant differences but had a lower ES, suggesting minor overestimation in matched conditions due to its higher ES and a closer $P$ value to significance. Finally, all other conditions showed significance.

Outliers and the whisker ranges of box plots for both ROIs in Figure 2 show variability within and between image types, with synthetic images exhibiting a broader area size range than real ones. Closer analysis shows more outliers in real images for lateral ventricles and in synthetic images for the hippocampus. Notably, 0.82% of synthetic ventricles' areas were below the minimum observed areas in real data. These sizes are clinically significant, indicating implausibly small lateral ventricles for adults (Agreed on by R1 & R2). Figure 3 b) illustrates such a case, clearly showing the differences in the unusual ventricle sizes of synthetic vs. real images for the same covariates, highlighting the need for rigorous evaluation of synthetic image generation for clinical use. Despite the segmentation model's training on real data, the model accurately segmented the unusually small ventricles (unseen in training), demonstrating robustness.

## 4. Discussion

The VTT reveals the challenge radiologists face in distinguishing synthetic from real brain MRI, attributed to the synthetic images' compelling realism and resulting in varied accuracy values. R2's frequent misclassifications, driven by cognitive biases, hint that performance could potentially improve through retraining with varied image types. Nonetheless, the smoothness effect is not only present in synthetic images but also in real ADNI and UKB images. Additionally, R2 consistently reported low confidence, often citing 'anatomical inaccuracies' in real images identified as synthetic, emphasizing the complexity of identifying subtle changes in ROIs linked to normal aging or AD progression. Although both radiol-

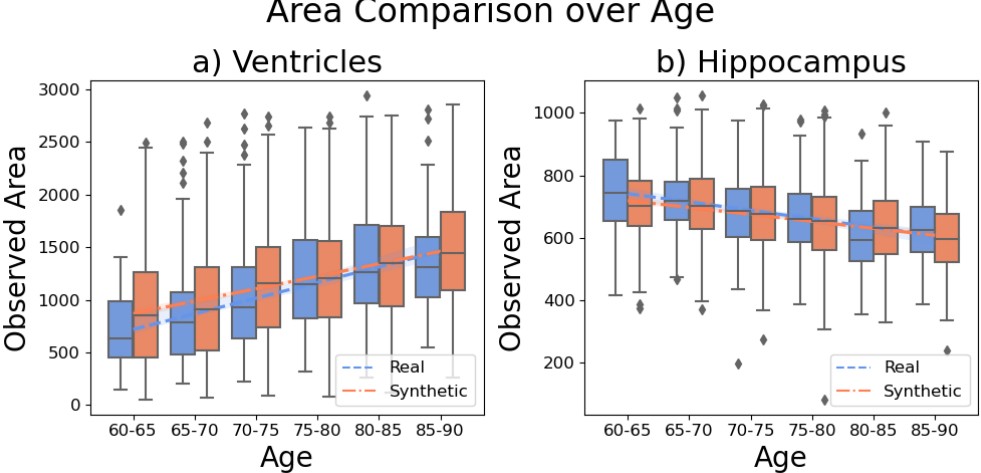

Figure 2: Age-wise comparison of observed LV and HC areas in real versus synthetic brain MRI across matched dists. shows the aging effect on brain ROI sizes.

ogists agreed and correctly identified the unusually small ventricles, manually inspecting 600,000 synthetic images to find similar conditions is infeasible. In contrast, our method allows for efficient identification of such cases. Based on these challenges, VTT alone is insufficient as a definitive measure of image realism, necessitating a more objective approach.

Our study advances this objective by rigorously evaluating the generative model's ability to replicate age-related anatomical changes, revealing its strengths and pinpointing limitations. Unlike the VTT, our method provides precise insights into anatomical accuracy, especially highlighting how the model overestimates ventricular enlargement while performing better in simulating hippocampal changes. This nuanced understanding highlights the importance of employing focused methods to evaluate the anatomical consistency of synthetic medical imagery. Finally, while observing a wide range of area sizes in synthetic images indicates no mode collapse, but outliers, particularly implausibly small ventricles, illustrates the model's current limitations in capturing the full spectrum of individual variability in real-world data.

The presence of biases in synthetic data, especially those that do not accurately reflect real-world conditions, poses a significant challenge in research on aging and AD diseases. When synthetic images inaccurately represent brain ventricles as larger than expected, this introduces an unrealistic bias that can mislead studies, potentially leading to incorrect conclusions about ventricular changes. This issue is compounded in areas like AD research, where patient data may be scarce, yet it is vital that the data used reflect genuine conditions to maintain the integrity of findings. The risk extends to diagnostic accuracy, where reliance on biased synthetic images could lead to misdiagnoses or the formulation of ineffective treatment plans. Similarly, training AI systems with inaccurate images could lead to errors in clinical applications, particularly those dependent on precise identification of age-related changes. To mitigate these challenges, it is crucial to refine generative models to more

faithfully represent the complexities of aging. By tackling these biases head-on, we can enhance the realism and reliability of synthetic images, thereby supporting more accurate clinical research, diagnostics, and the effective use of AI in healthcare settings.

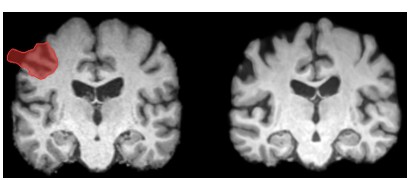

a) Real (left) vs. Synthetic (right) (Male, 66, CN).

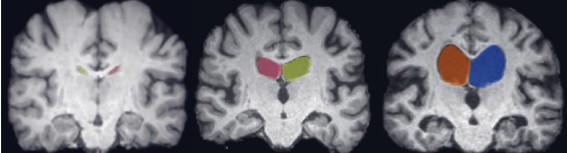

b) Synthetic (left) vs. Real (middle & right) (Female, 81, MCI).

Figure 3: a) R2's misclassification and incorrectly noting "anatomical inaccuracies" (Red brush) in a real image (Krichevsky, 2023), b) Unusually small LV sizes vs. real counterparts with identical covariates. Colored LVs represent our segmentation model's masks.

These discrepancies may stem from training with the large UKB dataset and fine-tuning on the smaller ADNI dataset. A potential direction is integrating UKB's demographic diversity with ADNI's AD-specific features, potentially improving the synthetic representation of underrepresented conditions. Additionally, advanced techniques could balance the influence of each data source, ensuring both cohorts' unique characteristics are captured in the generated images. Moreover, while causal models account for ROI volumes, the generative model's 2D training might restrict performance because of reduced contextual and spatial information. This limitation highlights the necessity for additional research as we evolve our 2D proof of concept to a more clinically relevant 3D model in future work.

## 5. Conclusion and Future Work

In conclusion, our study highlights the difficulties in differentiating synthetic from real brain MRI, emphasizing the limitations of the VTT and advocating for more objective evaluation methods. Our analysis reveals the generative model's strength in replicating age-related anatomical changes and simulating neurodegenerative features, alongside a tendency to introduce unrealistic biases, such as overestimated ventricular enlargement or implausibly small ventricles. These insights underline the necessity for robust evaluation methods for synthetic medical images, aiming to enhance image generation and facilitate their successful application in clinical settings.

While our evaluation primarily focuses on neurodegeneration, its relevance extends across various medical domains. For example, adapting our segmentation model to tumor analysis in different organs necessitates retraining for tumor segmentation. Our statistical pipeline would then be used to compare tumor shapes and sizes with those of actual tumors at various stages. In brain MRI studies, examining mid-line shifts and the progression effects of tumors on adjacent ROIs in both real and synthetic images offers a unique opportunity to discern consistencies and discrepancies between image types. Developing these evaluation methods, though challenging, is essential for the effective integration of synthetic images into clinical practice and enhancing the transparency of deep learning in such a critical field.

## Acknowledgments

The authors thank ImFusion GmbH for providing the software used to conduct the visual Turing test free of charge. We also acknowledge the UK Biobank Resource under Application ID Number 77717. Additionally, we thank the Alzheimer's Disease Neuroimaging Initiative (ADNI) for their resources.

Data used in preparation of this article were obtained from the Alzheimer's Disease Neuroimaging Initiative (ADNI) database (adni.loni.usc.edu). As such, the investigators within the ADNI contributed to the design and implementation of ADNI and/or provided data but did not participate in analysis or writing of this report. A complete listing of ADNI investigators can be found at: ADNI Acknowledgement List. Data collection and sharing for the Alzheimer's Disease Neuroimaging Initiative (ADNI) is funded by the National Institute on Aging (National Institutes of Health Grant U19 AG024904). The grantee organization is the Northern California Institute for Research and Education. In the past, ADNI has also received funding from various organizations listed in: ADNI Acknowledgement Section.

## Authors' contributions

HY and JF conceptualized the study. HY implemented the segmentation-based method in Python, conducted the analyses, and drafted the initial manuscript. JF assisted HY in interpreting the results and editing the script. WL developed the generative model and synthesized the brain MRI. AK prepared and conducted the Turing test and analyzed the results from that section with the help and supervision of JF and HY. Katrin Schanack and Beate Endemann are the radiologists who performed the Turing test. AR managed the preliminary processing of the datasets and obtained the segmentation ground truths. CL supervised the study. All authors contributed to the manuscript and approved its final version.

## Funding

This research was funded by the German Federal Ministry of Education and Research (BMBF) within the project 'Syreal' (Grant No. 01/S21069A). The funders had no role in the study design, data analysis, interpretation, or report writing.

## Data Availability

The ADNI dataset analyzed during the current study is available at the Alzheimer's Disease Neuroimaging Initiative (ADNI) database (Access Data) upon consenting to the data sharing agreement. Access to the UKB dataset used during the training of the segmentation models requires an application and is subject to approval (Register). The synthetic data used in the analysis are publicly available at figshare.

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

## Appendix A. Synthetic Data

Here, we provide more details on the conditional causal model, which generates the latent variables for the generative model. In this work, the synthetic images are generated from a conditional StyleGAN3 model with a handcrafted causal model, which controls the latent variables, *e.g.*, age, sex, and lateral ventricle volumes. Figure 4 depicts the causal modeling of the conditional variables. Age and sex are the confounders of the CDR and volumes. Furthermore, we use Maximum Likelihood Estimation to fit the training data into a handcrafted parametric model for all latent variables. Age is modeled by a beta dist, sex is a Bernoulli dist, while CDR is sampled as a multi-class softmax regression, conditioned on age and sex. There are three classes in CDR, "0" means a healthy subject, "0.5" means a cognitively impaired subject, and "1" means a dementia patient. Lastly, we use a Gaussian mixture regression model, conditioned on age, sex, and CDR, to model the volumes.

Therefore, with the trained causal model, we can randomly sample latent variables from the dist of the training set (Pawlowski et al., 2020). Furthermore, we can also sample specific variables, *e.g.*, CDR and volumes, by conditioning on a specific age. After sampling the latent variables, we fit them into the conditional StyleGAN3 to generate images with specific characteristics.

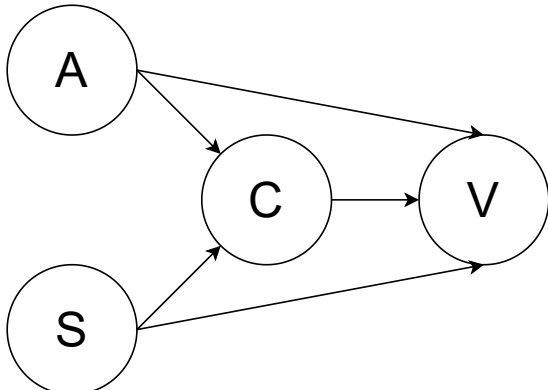

Figure 4: DAG (Directed acyclic graph) of the causal model. A: Age, S: Sex, C: CDR, V: Volumes (lateral ventricle volumes, cerebral cortex volumes, and hippocampus volumes.)

## Appendix B. Visual Turing Test (VTT)

Figure 5 shows the accuracy of two radiologists during the VTT, alongside the reasons cited for identifying synthetic images. Upon further analysis of the results, insights were gained into the certainty levels associated with the radiologists' decisions. R1 often indicated 'moderate certainty' in their choices, whereas R2 was more inclined towards 'very low certainty.' This variation in certainty directly correlates with their accuracy rates. Nonetheless, the task's complexity, from evaluating a single mid-brain slice (256x256) instead of a full 3D volume, which is atypical in clinical practice, posed significant challenges. In a subsequent meeting, the radiologists concurred that a repeat of the test, even after practicing, would unlikely improve their accuracy, highlighting the fundamental challenge of the task.

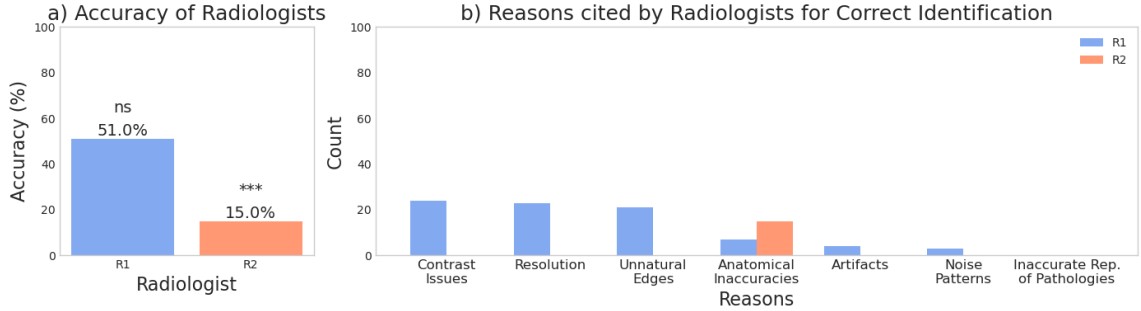

Figure 5: Bar plots of Radiologists' Accuracy values in VTT and Reasons for Correctly Identifying Synthetic Images.

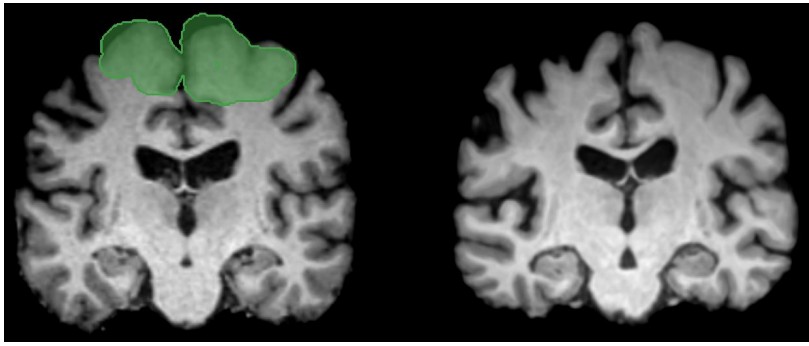

Figure 6: R1 misclassified a real image on the left as synthetic (actually on the right), stating "Noise Patterns" under "Other reasons" (green label brush) with no additional comments. The covariates assigned to both images indicate a male subject, 66 years old, and classified as CN (Krichevsky, 2023).

Additionally, we provide a breakdown of the reasons each radiologist cited for their choices, particularly when they correctly identified a synthetic image (see Figure 5 b)). R1 frequently attributed their decisions to 'contrast problems,' whereas R2, biased towards associating smooth images with real ones despite this characteristic being present in both image types, leaned toward 'anatomical inaccuracies.' as their rationale. Interestingly, neither radiologist selected "Inaccurate representation of pathology" as the reason for their choice. Lastly, Figures 3 and 6 demonstrate an example where both R1 and R2 misclassified a real image as synthetic for two different reasons ("anatomical inaccuracies" and "Noise Patterns," respectively).

## Appendix C. Segmentation Training and Evaluation

This appendix complements the main paper with detailed implementation, data-related specifics, and performance comparisons, which are crucial for reproducibility yet placed here to keep the main text focused.

### C.1. Model Architecture

We aimed to precisely segment ROI from brain MRI using state-of-the-art deep learning models, laying a solid foundation for evaluating age-related biological changes. We employed two principal architectures and their variations: **UNet** (Ronneberger et al., 2015) (both original and adapted versions) and **FCN with ResNet backbones (FCN)** (Long et al., 2015) (incorporating ResNet-50 and ResNet-101 variants). For the A-UNet, we increased the initial feature count to 64 from 32 and added an extra layer of depth, significantly boosting its segmentation efficiency for our dataset.

### C.2. Model Implementations

For consistency, we used standard optimizers and loss functions in A-UNet (lr = 0.0001, Adam optimizer with betas (0.9, 0.999), epsilon of 1e-8, dice loss) and FCN models (lr = 0.01, SGD with momentum of 0.9, weight decay of 1e-6, cross-entropy and Dice loss with auxiliary loss). We applied max normalization and augmentations (random rotations, flips). We customized models for the high-resolution UKB dataset and chose a batch size of 4 with gradient accumulation. Validation was conducted using a 5-fold cross-validation, dividing the UKB dataset into 60% training, 15% validation, and 25% testing. The training was performed exclusively on UKB to avoid bias towards real ADNI images selected as a reference for our evaluation approach.

We implemented our segmentation models on PyTorch Lightning and trained them on 1 A100 GPU until the convergence of the DSC score on a validation set. For consistency with synthetic images, we adapted to the 2D StylGAN3 pre-processing, registering MRI scans and ground truth masks to the MNI152 atlas space and resizing images to 256X256 using pytorch-complex package (Chatterjee et al., 2022) and extracting the central coronal 2D slice for training and evaluation.

### C.3. Models' Performance and Architectural Comparison

The DSC served as the primary metric to evaluate segmentation accuracy (Zou et al., 2004), reflecting the overlap between model predictions and SynthSeg ground truth masks. Initial optimization attempts started with models showing a DSC around 0.6, indicative of underfitting. Through iterative enhancements, the A-UNet model emerged as the top performer in internal testing on UKB and external testing on ADNI. The best-performing models were based on external testing, where A-UNet achieved the highest median DSC scores of 96.53% ± 0.16 for ventricles and 91.68% ± 0.28 for the hippocampus across 3 out of 5 folds (Table 2).

Comprehensive comparisons in Tables 2, 3 and 4 detail DSC scores across models and validation folds, illustrating A-UNet's superior performance and its selection for subsequent analyses.

Table 2: Comparison of model performance using **median DSC $\pm$ IQR of lateral ventricles and hippocampus ROIs** over 3 out of 5 folds cross-validation for the two best performing models, highlighting the highest obtained DSC values between both

| Model | Lateral Ventricles (DSC) | | Hippocampus (DSC) | |
|---|---|---|---|---|
| | Internal | External | Internal | External |
| **A-UNet** | **97.48% $\pm$ 0.04** | **96.53% $\pm$ 0.16** | **94.37% $\pm$ 0.04** | **91.68% $\pm$ 0.28** |
| FCN-Res101 | 94.20% $\pm$ 0.01 | 93.34% $\pm$ 0.42 | 91.62% $\pm$ 0.05 | 88.33% $\pm$ 0.27 |

Table 3: Comparison of model performance using **mean DSC of lateral ventricles and hippocampus ROIs** over the first fold in cross-validation for all models

| Model | Lateral Ventricles | | Hippocampus | |
|---|---|---|---|---|
| | Internal | External | Internal | External |
| **A-UNet** | **97.52%** | **96.68%** | **94.37%** | **91.61%** |
| UNet | 96.16% | 94.65% | 91.60% | 88.09% |
| FCN-Res101 | 94.21% | 93.71% | 91.45% | 86.96% |
| FCN-Res50 | 94.20% | 93.55% | 91.13% | 84.79% |

Table 4: Comparison of model performance using **median DSC of lateral ventricles and hippocampus ROIs** over the first fold in cross-validation for models trained on Intracranial ROI

| Model | Intracranial Areaa (DSC) | |
|---|---|---|
| | Internal | External |
| **A-UNet** | **99.37%** | **98.57%** |
| FCN-Res50 | 99.10% | 98.08% |

## C.4. Outlier Analysis

In our final analysis phase, we investigated extreme outliers within the lateral ventricles and hippocampus regions to detect potential segmentation inaccuracies. Notably, errors were confined to a handful of cases in both real and synthetic images, mainly resulting from the lack of typical T1-weighted contrast inherent to each image type. Furthermore, we encountered two empty synthetic images. Nonetheless, excluding these outliers from our

dataset did not influence the overall significance of our results. It appears that the observed irregular contrast patterns can be traced back to anomalies in the real dataset, which were then mirrored in some synthetic images, potentially arising from errors in scanning protocols. These discrepancies introduce segmentation challenges by creating contrast variations distinct from those in the training set. For visual examples of these anomalies and the segmentation issues they caused, see Figure 7.

| Image | Lateral Ventricles | Hippocampus | Intracranial |
|-------|--------------------|-------------|--------------|

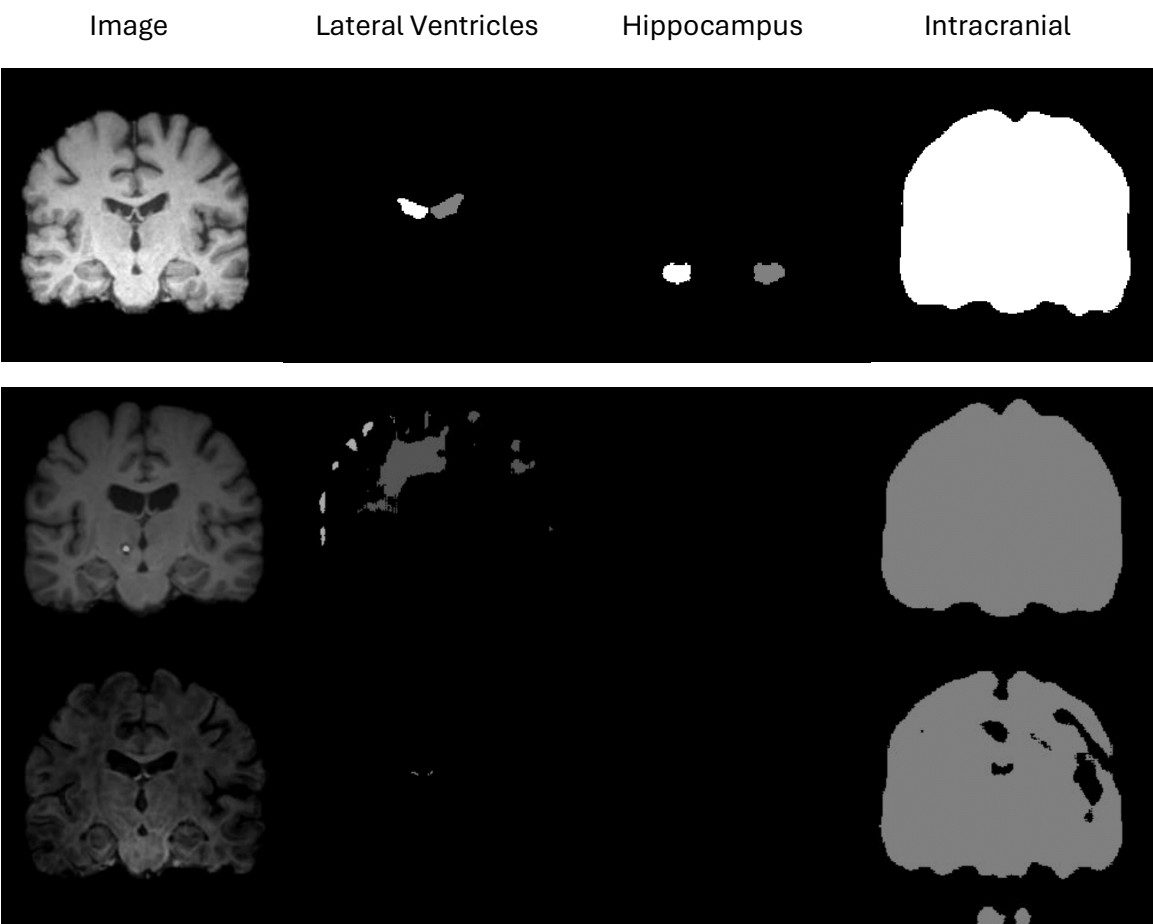

Figure 7: The first row displays the successful segmentation of three ROI masks in normal T1-weighted contrast used for training. In contrast, the second and third rows highlight segmentation challenges in contrasts different from training. The first column shows the image, followed by segmentation masks for lateral ventricles, hippocampus, and intracranial area. The first and second rows feature real images, while the third row shows a synthetic image.

## Appendix D. Supplementary Analyses on Biological Plausibility

### D.1. Results

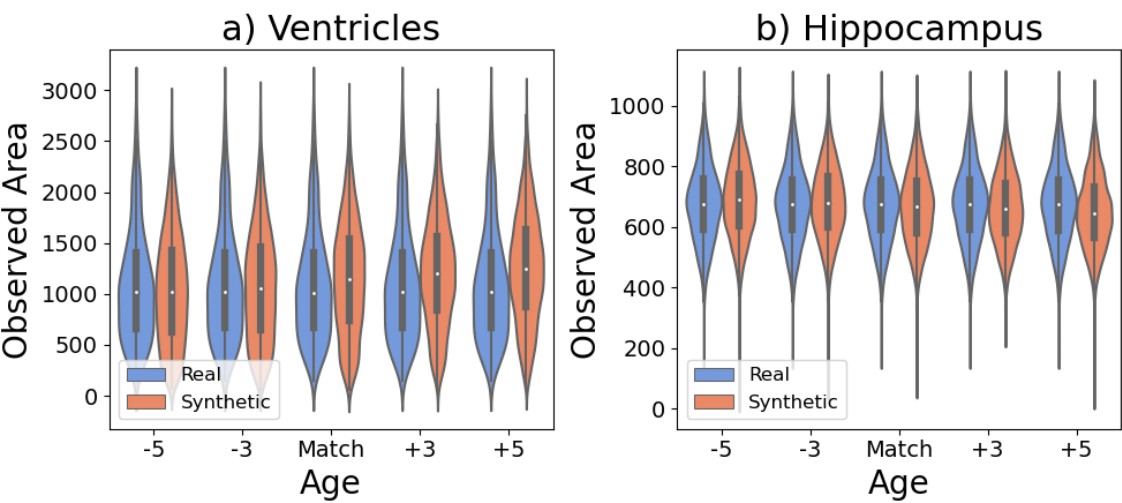

Figure 8: Comparison of ROI areas in real vs. synthetic brain MRI across different age dists: Matched dists. are centered, showcasing a median age of 74. On the sides, mismatched dists. where synthetic images have a higher or lower median age ($\pm$ 3 and 5 years).

Table 5: **Statistical Comparison of ROI Areas in Real vs. Synthetic Images Across different CDR dists:** Analysis examines ROI areas under **Matched Conditions** (covariates align, MCI predominant) and Shifted Conditions (Synthetic images' dist is changed to either CN or AD dominance). **Significant differences** are highlighted.

| ROI | Cond. | Permute | ES | $CI_l$ | $CI_u$ | KS |
|-----|-------|---------|-----|--------|--------|-----|
| LV | CN | 2.00E-01 | -4.80E-02 | -1.30E-01 | 2.90E-02 | **3.8e-05** |
| LV | Match | **2.0e-03** | -1.30E-01 | **-2.00E-01** | **-5.00E-02** | **2.0e-08** |
| LV | AD | **0.0e+00** | -1.90E-01 | **-2.70E-01** | **-1.10E-01** | **2.0e-12** |
| HC | CN | 7.00E-01 | 1.50E-02 | -6.20E-02 | 9.20E-02 | 2.20E-01 |
| HC | Match | 5.70E-02 | 7.00E-02 | -6.70E-03 | 1.50E-01 | 8.70E-02 |
| HC | AD | **0.0e+00** | 2.10E-01 | **1.30E-01** | **2.80E-01** | **7.5e-08** |

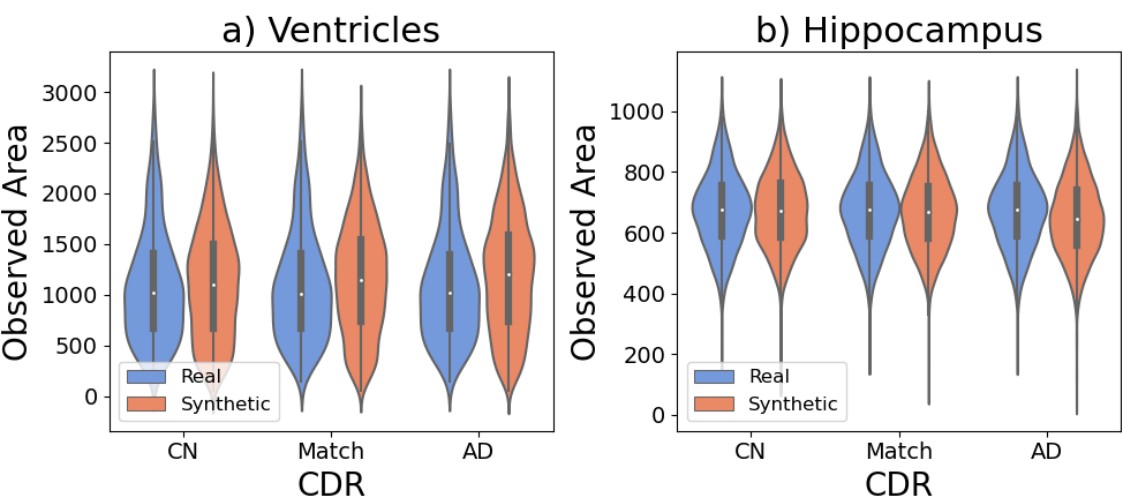

Figure 9: Comparison of ROI areas in real vs. synthetic brain MRI across different CDR dists: Matched dists. are centered, showcasing aligned covariates with MCI predominance. On the sides, mismatched dists. show synthetic images with CN predominance on the left and AD predominance on the right.

Table 6: Statistical Comparison of ROI Areas in Real vs. Synthetic Images across Different Sex dists: Analysis examines ROI areas under **Matched Conditions** (Approx. balanced Female to male ratio) and Shifted Conditions (Synthetic images either female or male predominant at 40% and 60%). **Significant differences** are highlighted.

| ROI | Cond. | Permute | ES | $CI_l$ | $CI_u$ | KS |
|-----|-------|---------|-----|--------|--------|-----|
| LV | Females | 2.10E-01 | -5.00E-02 | -1.30E-01 | 2.70E-02 | **1.7e-04** |
| LV | Match | **2.0e-03** | -1.30E-01 | **-2.00E-01** | **-5.00E-02** | **2.0e-08** |
| LV | Males | **1.2e-02** | -9.70E-02 | **-1.70E-01** | **-2.00E-02** | **1.2e-07** |
| HC | Females | 3.70E-01 | -3.60E-02 | -1.10E-01 | 4.10E-02 | 2.70E-01 |
| HC | Match | 5.70E-02 | 7.00E-02 | -6.70E-03 | 1.50E-01 | 8.70E-02 |
| HC | Males | 2.10E-01 | 4.70E-02 | -3.00E-02 | 1.20E-01 | 1.30E-01 |

The insights derived from Table 5, and Table 6 are pivotal in comparing synthetic and real data dists. We see a similar pattern to what was observed in hippocampus in age mismatches, where dists. predominantly featuring Cognitive Normal (CN) or Females also have lower non significant permute $P$ values than matched dists, which have $P$ values close to

Figure 10: Comparison of ROI areas in real vs. synthetic brain MRI across different sex dists: Matched dists. are centered, showcasing a Approx. balanced female-to-male ratio. On the sides, show synthetic images with a female predominance (60%) on the left side and a male predominance (60%) on the right side.

significance. Furthermore, in lateral ventricles, CN- or female-predominated dists.' permute $P$ values are non-significant, in contrast to matched, and hence more closely resemble real data than matched distributions (MCI predominant, balanced sex ratio). In Figure 9, and Figure 10, violin plots are more similar in the hippocampus than in the lateral ventricles, reflecting the non-significant KS $P$-values for the hippocampus in contrast with the lateral ventricles. The only significant KS and shape differences in the hippocampus occur in the AD-predominant dist mismatch. These results align with the age mismatch findings and suggest that certain demographic characteristics influence the model's ability to generate synthetic images that accurately reflect the diversity of real anatomical structures.

### D.2. Discussion

The observed results highlight the significance of incorporating covariates in the training and generating synthetic medical images. Our approach, utilizing a causal model that accounts for these covariates, aims to address this need. The increased resemblance of CN-predominated dists. to real data suggests that our synthetic models may be more adept at capturing the anatomical nuances of cognitively normal subjects, possibly due to initial training on extensive assumed to be healthy (CN) UKB cohort or reflecting the varied anatomical characteristics associated with neurodegenerative conditions like MCI and AD. The closer alignment of female-predominated dists. with real data further prompts a review of the model's sensitivity to sex-specific anatomical differences.

### D.3. Future Work

Future research should explore the underlying mechanisms contributing to these observed discrepancies in synthetic image generation. It is essential to investigate the model's training data and algorithms for potential biases or limitations in capturing the full spectrum of human anatomical diversity. Additionally, expanding the model's training dataset to include a more diverse representation of ages, cognitive states, and sexes may help generate synthetic images that more accurately mirror the variability found in real-world data.

