# OpenReview forum: "Evaluating Age-Related Anatomical Consistency in Synthetic Brain MRI against Real-World Alzheimer's Disease Data."
_MIDL.io/2024/Conference — MIDL 2024 Oral_

### Official Review · Reviewer_Cavr · 2024-02-26

**Confidence:** 5
**Preliminary Rating:** 2
**Final Rating:** 4

**Summary:**

The paper proposes a full pipeline to 1) generate synthetic brain images, 2) segment the synthesized images and 3) compute several metrics to assess their quality. The authors train a generative model and generate images conditioned on a set of factors (age, sex, cortex volume, etc.) matching real images from the ADNI database. Then, they used a U-Net (trained on UKBiobank data) to segment brain structures from the real and synthetic images. Finally, the extract the volume of different brain structures in the real and synthetic images and compare them.

**Strengths:**

The paper contains a lot of information and proposes a full pipeline from conditioned synthetic image generation to image quality evaluation. A lot of methods are mentioned. The scope of the paper is relevant for MIDL.

**Weaknesses:**

Along the paper, the main point of interest of the paper seems to shift and I could not understand the novelty of the proposed approach.
From the abstract it seems that the authors want to emphasize that they propose novel metrics for the evaluation of synthesized brain images. They state that they use existing methods for image generation and segmentation (“followed by applying a U-Net model”), but later on the attention is focused on the development of a novel segmentation model.

In the end, it is not clear what is being tested and compared. Does the observed differences between the real and synthesized data come from the generative model or from the segmentation model? In the current state of the paper, it is not possible to answer that question as nor the generative model neither the segmentation model are thoroughly tested and compared to other existing methods.

If the main point of the paper is to propose a novel segmentation method then the introduction should focus on existing segmentation methods (QuickNat [1], FastSurfer [2], SynthSeg [3]).

Then the proposed method should be described in the main part of the paper, not in Appendix.
Finally, the method’s results should be compared with state of the art models [1-3] and segmentation results should be displayed in the main body of the paper.

If the main point of the paper is to propose new metrics for assessing the quality of synthesized images, then it should cite relevant literature in which some of the proposed metrics have already been applied (Riberio et al. [4]-> already looked at ventricular volume to assess image quality, Wilms et al. [5])

[1] Roy, A. G., Conjeti, S., Navab, N., Wachinger, C., & Alzheimer's Disease Neuroimaging Initiative. (2019). QuickNAT: A fully convolutional network for quick and accurate segmentation of neuroanatomy. NeuroImage, 186, 713-727.
[2] Henschel, L., Conjeti, S., Estrada, S., Diers, K., Fischl, B., & Reuter, M. (2020). Fastsurfer-a fast and accurate deep learning based neuroimaging pipeline. NeuroImage, 219, 117012.
[3] Benjamin Billot, Douglas N. Greve, Oula Puonti, Axel Thielscher, Koen Van Leemput,
Bruce Fischl, Adrian V. Dalca, and Juan Eugenio Iglesias. Synthseg: Segmentation
of brain MRI scans of any contrast and resolution without retraining. Medical Image
Analysis, 86:102789, 2023. ISSN 1361-8415. doi: 10.1016/j.media.2023.102789.
[4] F. D. S. Ribeiro, T. Xia, M. Monteiro, N. Pawlowski, B. Glocker, in Proceedings of the 40th International Conference on Machine Learning (PMLR, 2023; https://proceedings.mlr.press/v202/de-sousa-ribeiro23a.html), pp. 7390–7425.
[5] M. Wilms et al., IEEE Transactions on Medical Imaging. 41, 2331–2347 (2022)

**Detailed Comments:**

Conclusion: “In conclusion, the study presents a model that offers significant promise in replicating age-related anatomical changes, particularly for the hippocampus.” I disagree with that sentence. The beginning of the paper says that a segmentation model is proposed but that the generative model is based on existing work. If the main point of the paper is to propose a novel generative model, then appropriate literature should be cited, the model should be described in details with a graphical description and other generative approaches should be compared.

**Justification Of Final Rating:**

Thank you to the authors that did a considerable amount of modifications to the paper. I updated my rating regarding that. The goal of the paper is now clearer and I believe tackles an important topic (results evaluation for gerenerative models), although not proposing very novel methods. I noticed a few typos remaining (see my last comment), I would advice the authors to read through the paper carefully one more time before it's potential presentation in MIDL.

**Justification Of The Preliminary Rating:**

The goal of the paper is not clear and it contains too much information. It is also not clear if the authors propose a novel method or validate existing methods. I believe the paper should focus on one step of the pipeline, contain a literature review focused on that step and clearly state what the works brings to the community.

**Questions To Address In The Rebuttal:**

The paper needs major rewriting to clearly explain what it presents: a novel image generation algorithm? A novel segmentation method? Novel metrics for assessing the quality of synthetic images?

The literature review of recent work should be appropriate and coherent regarding the paper main goal.

The results should be compared to SOTA methods regarding the main goal of the paper, unless the paper can be considered as a method paper if the innovation on the generation of segmentation method is clearly explained.

**Special Issue:**

No

---

> ### Author Response · Authors · 2024-03-18
> **Response to Reviewer Cavr**
>
> Thank you for your constructive feedback. Based on your comments, as well as insights from others, I have extensively revised the paper. This process has been an invaluable learning opportunity and I hope you reconsider your initial rating based on the enhanced manuscript.
>
> ## Questions To Address In The Rebuttal:
>
> >The paper needs major rewriting to clearly explain what it presents: a novel image generation algorithm? A novel segmentation method? Novel metrics for assessing the quality of synthetic images?
>
> Initially, we faced challenges in articulating our thoughts clearly in the paper, given its multifaceted nature. (The main goal is to develop novel metrics for assessing the quality of synthetic images. While the generative and segmentation methods were utilized as tools, they significantly impacted the work and understanding their roles are paramount ). However, the constructive guidance received has enabled me to significantly revise the manuscript. I undertook major rewrites to closely integrate all comments provided by the reviewers.
>
> >If the main point of the paper is to propose a novel segmentation method then the introduction should focus on existing segmentation methods (QuickNat [1], FastSurfer [2], SynthSeg [3]).
>
> Developing accurate segmentation models was essential, though not the main focus of our study. Given that our generative model is currently a proof-of-concept limited to 2D, with future plans to expand to 3D, we found that well-established methods like SynthSeg, designed for 3D, were not directly applicable. As a workaround, we used SynthSeg to establish ground truth data. Attempts to implement QuickNat faced challenges; it underperformed due to its reliance on auxiliary variables from SynthSeg and the need for fine-tuning with expert-annotated data—resources we did not have access to (Not included in Appendix, no beneficial value to our work). FCN-Res, which is optimized for 2D images, was identified as the second-best option and is further discussed in the appendix.
>
> >The literature review of recent work should be appropriate and coherent regarding the paper main goal.
>
> Yes, we agree and we have expanded on specific literature on evaluation methods you mentioned (Riberio et al. and Wilms et al.) and more in the intro.
>
> >The results should be compared to SOTA methods regarding the main goal of the paper, unless the paper can be considered as a method paper if the innovation on the generation of segmentation method is clearly explained.
>
> We initially aimed to compare our work with the state-of-the-art (SOTA), particularly in terms of the Visual Turing Test (VTT), but acknowledge that our writing may not have clearly conveyed this intention. Understanding this, we have thoroughly revised the paper to articulate the main goal more clearly and explain why VTT represents our chosen SOTA. This clarification, supported by a literature review in the introduction, explains why VTT is most suitable for our research objectives.

---

> > ### Comment · Reviewer_Cavr · 2024-03-27
> > **Few final corrections**
> >
> > By reading the new version of the paper, I have noticed a few typos that remains (see some of them below):
> > Introduction paragraph 3: Should start with capital letter (visual Turing test)
> > Figure 1 arrows going to the bottom boxes are cut
> > Section 2.1 paragraph 1 – problem with the first line
> > Section 2.1 paragraph 2 line 4: Ours should be ours

---

> > > ### Author Response · Authors · 2024-03-27
> > > **Replying to Few final corrections**
> > >
> > > Thank you for your meticulous attention to detail. I have addressed all the suggestions and am actively reviewing the entire text for any additional issues. Is there anything else we need to cover? Have we thoroughly answered all of your questions, or is there anything more? After reviewing the revised version, did you perhaps reconsider your initial rating?
> > >
> > > Thank you once again for your valuable time and efforts.

---

### Official Review · Reviewer_XzGi · 2024-02-28

**Confidence:** 4
**Preliminary Rating:** 5
**Recommendation:** Oral

**Summary:**

The authors use a generative model to generate images of the ageing brain with and without AD. They assess the age-related and disease-induced changes in the real data and compare this to the visible changes in the generated images using first a visual assessment and secondly quantitative metrics based on the volumes of segmentations of ventricle areas. They find that radiologists can be fooled by the generated images, but that volumetry of the ventricles shows slight to significant differences in the metrics assessed. They conclude that visual inspection alone isn't conclusive about the quality of generative models, but that metrics are needed that are objective and ideally causal.

**Strengths:**

The paper is well structured, clearly written, well motivated and mostly sound from the research perspective. It presents a short overview motivating the research into metrics for quantitative assessment of generative models.

The authors then present their methodolgy including the data in a clear manner leaving close to no questions open.

The same holds for the presentation of the results; they are clearly written, show figures that underline the main findings of the study properly and are explained in enough depth to critically acclaim the quality of the generative model.

In the conclusion, the authors point out the lessons learned: VTT alone doesn't suffice to judge the quality of a generative model. Metrics are needed. Further research is needed what the model actually "learns" so that it represents certain factors well, and other less so.

**Weaknesses:**

You state that you set aside data for external testing to prevent bias, but then say that after iterative refinement, you arrive at the best-performing model on the external test set. This sounds like you applied intermediate models on the test set, which isn't a proper thing to do in my opinion. Perhaps the description in this section 2.4 is just too short to convey the actual procedure. In any case, this should be clarified. It is of lesser concern as it only regards the segmentation model, which in turn is only used to measure the volumes, and if it has no bias towards any of the data subsets, it will serve its purpose even if the test set was tainted through repeated use for model selection.

**Detailed Comments:**

I was torn about the length of the appendices. I had no second paper in my review that deferred so much material into the appendix. On the one hand, this is an indication of the thoroughness of the research, on the other hand, I find it hard to deal with in the short amount of time reviewers have for the reviews assigned to them. I have decided not to read any of the appendices of this or any other submission and base my evaluation solely on the main sections.

In 2.2.2, you speak of causal controls of labels by changing age and continue to say "since volumes are correlated with age". Strange to read about correlation here -- isn't age causing the volume change (even if through mediators and influenced by possible confounders, and therefore only indirectly)?

In 2.5.1 you refer to a "unified dataframe" -- what's that?

I was puzzled to read about the sub-chance performance of the second radiologist in the VTT. It seems that he or she was perfectly able to tell the two image categories apart, just "mixed up" the categories for whichever reason. I would have liked to hear what his or her criteria were -- perhaps the generated images looked more pleasing or something? In any case, there might have been a confounder that this observer was able to latch onto, and I would have had the desire to pinpoint this.

**Justification Of The Preliminary Rating:**

This contribution was quite inspiring to read. There were many sections where I would have liked to talk with the authors and enter a discussion. I could imagine that this will be the case for the MIDL audience as well, and that's my motivation to suggest it for oral presentation.

**Questions To Address In The Rebuttal:**

All above "weaknesses" and "detailed comments"...!

**Special Issue:**

No

---

> ### Author Response · Authors · 2024-03-18
> **Response to Reviewer XzGi**
>
> Thank you for your valuable review. As this is my first paper, I was particularly eager for it to be understood in the manner you have described. Your feedback, along with that from other reviewers, has been instrumental in my learning process, significantly improving the manuscript's quality. We all appreciate the opportunity to refine my work based on such constructive guidance and hope that all your comments have been addressed satisfactorily.
>
> ## Weaknesses:
> > You state that you set aside data for external testing to prevent bias, but then say that after iterative refinement, you arrive at the best-performing model on the external test set. This sounds like you applied intermediate models on the test set, which isn't a proper thing to do in my opinion. Perhaps the description in this section 2.4 is just too short to convey the actual procedure. In any case, this should be clarified. It is of lesser concern as it only regards the segmentation model, which in turn is only used to measure the volumes, and if it has no bias towards any of the data subsets, it will serve its purpose even if the test set was tainted through repeated use for model selection.
>
> Yes, it was too short and indeed confusing. Therefore, we amended it in Sec. 2 under "Quantification of ROIs". ( Trained and validated solely on UKB for peak performance in internal testing. A-UNet excelled in UKB internal and ADNI external tests. Had A-UNet excelled on UKB internally and FCNResNet externally on ADNI, I would have chosen FCNRes to enhance performance on ADNI, my key dataset for analysis.)
>
>
> ## Detailed Comments:
> > I was torn about the length of the appendices. I had no second paper in my review that deferred so much material into the appendix. On the one hand, this is an indication of the thoroughness of the research, on the other hand, I find it hard to deal with in the short amount of time reviewers have for the reviews assigned to them. I have decided not to read any of the appendices of this or any other submission and base my evaluation solely on the main sections.
>
> I understand and appreciate your time and effort. Yes, thoroughness is essential for the research, especially given the multifaceted nature of this work.
>
> > In 2.2.2, you speak of causal controls of labels by changing age and continue to say "since volumes are correlated with age". Strange to read about correlation here -- isn't age causing the volume change (even if through mediators and influenced by possible confounders, and therefore only indirectly)?
>
> Couldn't agree more, this was an oversight from our side; it should have been 'causal' and we corrected it.
>
> >In 2.5.1 you refer to a "unified dataframe" -- what's that?
>
> python dataframe of all the 5:1 samples in one tabular data for analysis purposes. Thank you for highlighting this. We've clarified this approach in the revised Sampling subsection, now located in Section 3 under 'Experiment Setup'
>
>
> >I was puzzled to read about the sub-chance performance of the second radiologist in the VTT. It seems that he or she was perfectly able to tell the two image categories apart, just "mixed up" the categories for whichever reason. I would have liked to hear what his or her criteria were -- perhaps the generated images looked more pleasing or something? In any case, there might have been a confounder that this observer was able to latch onto, and I would have had the desire to pinpoint this.
>
> Yes, we were confounded by space and might not have fully indulged in this aspect before, we expanded on that in the Discussion sec. 4 1st paragraph.

---

### Official Review · Reviewer_cJk8 · 2024-03-05

**Confidence:** 3
**Preliminary Rating:** 3
**Final Rating:** 4

**Summary:**

The key idea is to propose a method to evaluate the realism of generative models. Particularly the paper focus on T1 brain generation of T1 based on multiple inputs. The claim of the paper is Visual Turing Test is not enough to evaluate the quality of image generation especially the fidelity. The authors propose an other analysis based on the ventricule segmentation and the hypocampus segmentation size. The segmentation is trained by a U-Net using synthseg segmentation as ground truth. It shows a bias of generation based on the age (with a larger median area for the ventricles).

**Strengths:**

- Showing the weakness of a widespread practice which is the Visual Turing Test;
- In the context of alzheimer desease, the paper illustrate a bias of StyleGanV3 therefore, the generative model generate some impossible cases.

**Weaknesses:**

In the content:

- Only StyleGAN v3 has been evaluated It would have been much better to also evaluate diffusion models for example the author could have use [1, 2, 3] or any other model.

-"Despite these challenges, deep generative models like generative adversarial networks
(GANs) and Diffusion models are advancing synthetic medical image generation."  For this sentence in the introduction, it would have been better to cite GAN and diffusion model  (in medical images and natural images).

- The analysis based on the segmentation is rather biased. The segmentation and the generation are on 2D slices. The bias on the ventricle could come from the bias that some slice are less sampled even if the authors specify that only the central slice is generated and the pixel counting is divided by the cranial size. How do the authors take into account the view of the 3D model. This analysis would be more relevant for 3D generation.

- "These findings, especially the smaller synthetic ventricles that fall below the minimum area value of real data, bear clinical significance and suggest an implausible size for an adult, warranting further investigation into the model’s training and generative processes."  Does the author have an explanation why  Is it because of the generative model algorithm, the lack of data ? This is why In the reviewer's opinion, it is important to evaluate different generative models (as said above) ?

-The appendix B4 experiment is rather weird. The ground truth is synthseg and the methods are comparison with synthseg... Because we have synthseg, why do we need to train a model  and do not use directly synthseg ?

In the form;
In general, there is a lack of illustration.
- lack of a illustration at the beginning to summarize the protocol.
- It is a paper about generative models, yet we do not see in the paper a lot of generated brain. It would have been great to show examples of misclassification of the expert for example In the paper (The figure).
- The figure 5 in the appendix is not very clear, subtitles are missing next to the images. Is it the result of segmentation model? Do we see a failure of the segmentation model

- An explanation of the causal model would have been great (at least in the appendix). It is not clear based on the paper how it works

[1] https://github.com/FirasGit/medicaldiffusion
[2] https://github.com/Stability-AI/generative-models
[3] Pinaya, Walter HL, et al. "Brain imaging generation with latent diffusion models." MICCAI Workshop on Deep Generative Models. Cham: Springer Nature Switzerland, 2022.

**Detailed Comments:**

Please see the weaknesses for minor improvement.
I would insist on adding a general figure to correctly summarize the evaluation protocol.

**Justification Of Final Rating:**

First, the reviewer wants to thank the authors for the interesting conversation.
The reviewer understand that the goal of this paper is not to bring a new image generation or a new segmentation algorithm but to bring a new evaluation protocol for synthetic image generation suited for Alzheimer.
The reviewer understand the choice of the authors for 2D generation. 3D generation is much more challenging than 2D generation.
The reviewer still thinks that the 2D generation is an important limitation and the lack of comparison with other SoTA model also.

But the reviewer still acknowledges the amount of work done by the authors, and the importance of this research. Also thanks to the revision, the paper is also clearer.
The reviewer has still uncertainty in the final rating but the authors and the reviewer XzGi have convinced the reviewer of changing its mark.

**Justification Of The Preliminary Rating:**

Borderline, but the reviewer rather tends to weak reject because of the weakness mentionned above. The final rating will depend on how the authors answer well in the rebuttal and the review of the other reviewers.

**Questions To Address In The Rebuttal:**

See weakness section:
Some question

- Does 2D generation biases the analysis because of the lack of context, and we need 3D generation ?
- Why did the author test only StyleGanv3 ?
- The evaluation protocol is maybe too specific to Alzheimer. How to extend the protocol to other cases ?
-  Based on the experiments the authors made, What are the possible improvements to generate better synthetic MRI ?

**Special Issue:**

No

---

> ### Author Response · Authors · 2024-03-18
> **Response to Reviewer cJk8**
>
> Thank you for your valuable feedback and your time. We appreciate your eyes for details and your constructive criticism that helped further enhance the manuscript. We tried our best to accommodate your comments. If these clarifications address your concerns, we kindly ask that you reconsider your initial rating.
>
> ## Questions To Address In The Rebuttal:
> ### In the content:
>
> > Only StyleGAN v3 has been evaluated It would have been much better to also evaluate diffusion models for example the author could have use [1, 2, 3] or any other model. Why did the author test only StyleGanv3 ?
>
> Thank you for raising this point; the thought has crossed my mind. However, the main goal (evaluation) was not initially clear. We have amended this in the new PDF, where we expanded on evaluation-specific literature in the introduction and demonstrated that, unlike current studies, our purpose is not to compare generative models (state-of-the-art, SOTA) but to provide a comprehensive evaluation of the anatomical consistency of synthetic brain MR images, specifically in the context of aging and Alzheimer's disease (AD), against real-world data. Additionally, there is no available public synthetic AD dataset or checkpoint to reproduce the Khader et al. (2022) paper, deeming it infeasible.
>
> >"Despite these challenges, deep generative models like generative adversarial networks (GANs) and Diffusion models are advancing synthetic medical image generation." For this sentence in the introduction, it would have been better to cite GAN and diffusion model (in medical images and natural images).
>
> Thank you for pointing it out, we added references in intro.
>
> >The analysis based on the segmentation is rather biased. The segmentation and the generation are on 2D slices. The bias on the ventricle could come from the bias that some slice are less sampled even if the authors specify that only the central slice is generated and the pixel counting is divided by the cranial size. How do the authors take into account the view of the 3D model. This analysis would be more relevant for 3D generation. Does 2D generation biases the analysis because of the lack of context, and we need 3D generation ?
>
> It could be influenced by the lack of context and we acknowledged that in discussion section. On the other hand, experts' difficulty in distinguishing between real and synthetic images, coupled with our analysis showing good performance for the hippocampus but inconsistencies in lateral ventricles for certain subgroups, suggests that the generative model holds promise but also has areas for improvement. I have incorporated a more detailed discussion on a probable source of bias, informed by our findings, in the last paragraph of the Discussion section (Section 4) in the updated PDF. The current generative adversarial network (GAN) model serves as a proof of concept, and future plans include advancing towards 3D image generation, which is more relevant and potentially more impactful for clinical applications.
>
>
> >"These findings, especially the smaller synthetic ventricles that fall below the minimum area value of real data, bear clinical significance and suggest an implausible size for an adult, warranting further investigation into the model’s training and generative processes." Does the author have an explanation why Is it because of the generative model algorithm, the lack of data ? This is why In the reviewer's opinion, it is important to evaluate different generative models (as said above) ?
>
> This is an important point and we expanded on potential bias reason in disc. sec 4 last paragraph. Importance of evaluating different generative models (answered above) and added to the expanded literature review in intro.
>
> >-The appendix B4 experiment is rather weird. The ground truth is synthseg and the methods are comparison with synthseg... Because we have synthseg, why do we need to train a model and do not use directly synthseg ?
>
> I rewrote parts of the scripts to make this clearer. Synthseg as well as other established statistical methods like FSL and FreeSurfer are limited to 3D and we are limited to 2D synthetic images (See sec 2.3 under ROI Quantification). It was used to generate segmentation masks to train our 2.5D approach capable of both 2 and 3D
>
> ### In the form; In general, there is a lack of illustration.
>
> > lack of a illustration at the beginning to summarize the protocol.
>
> True, we had one but couldn't fit it before. However,  we added a diagram under methodology sec 2 in the new pdf
>
> >It is a paper about generative models, yet we do not see in the paper a lot of generated brain. It would have been great to show examples of misclassification of the expert for example In the paper (The figure).
>
>  We hear you and therefore added image illustration (now fig 3) of missclassification example and unusually small ventricles as well as another example in appendix.

---

> > ### Author Response · Authors · 2024-03-18
> > **Further Response to Reviewer cJk8**
> >
> > >The figure 5 in the appendix is not very clear, subtitles are missing next to the images. Is it the result of segmentation model? Do we see a failure of the segmentation model
> >
> > We can see how it could have been done better, thank you. Thus we fixed it by adding a successful images segmentation and extra annotations and explanations to make it clearer. (Segmentation failure happened in a couple of images (3-4) and excluding them as explained in appendix didn't effect our initial statistical results.)
> >
> > >An explanation of the causal model would have been great (at least in the appendix). It is not clear based on the paper how it works
> >
> > Yes we agree and we added more on it in appendix a as requested.
> >
> > >The evaluation protocol is maybe too specific to Alzheimer. How to extend the protocol to other cases ?
> >
> > We mentioned that it is applicable but didn't demonstrate how, hence we added future work regarding this in sec. 5 paragraph 2
> >
> > >Based on the experiments the authors made, What are the possible improvements to generate better synthetic MRI ?
> >
> > While identifying the weaknesses and strengths of the generative model to aid in the refinement of the generative models, is primarily our aim, we have included relevant content in the last paragraph of the of Disc. sec. 4.

---

### Author Response · Authors · 2024-03-18
**General Comment**

Thank you to all reviewers for your valuable feedback and the time you've dedicated. We have made a concentrated effort to incorporate your comments, enhancing the clarity and depth of our manuscript. If these revisions address your concerns, we kindly request you to reconsider your initial rating.

## Key Adjustments in the Rebuttal:

1. Focus on StyleGAN v3: Selected due to its significance for our study's objectives concerning Alzheimer's disease. In the revised manuscript, we've elaborated on this selection and clarified that comparing various state-of-the-art generative models isn't our priority, our emphasis is instead on providing a comprehensive evaluation of synthetic image quality and anatomical consistency.

2. Bias in 2D Slice Analysis: Addressed potential biases and acknowledged the need to move towards 3D generation for enhanced performance and clinical relevance, detailed in the Discussion section.

3. Visual Illustrations Enhanced: Added examples of misclassifications and anomalously small ventricles to better visualize our findings.

## Clarifications Made:

- Segmentation Method Choice: Justified our approach, emphasizing the constraints of applying 3D methods to our 2D work and opting for a suitable alternative.

- Causal Model and  Expanded in the Appendix

- Application to Other Fields: Discussed the protocol's extension to other medical fields.

- Future Directions for Synthetic MRI Generation: Outlined Areas for Improvement in Generative Model Refinement.

We appreciate the opportunity for refinement based on your constructive feedback and hope the manuscript now effectively addresses the raised points.

---

### Author Response · Authors · 2024-03-25
**A Kind Reminder**

Dear all reviewers,

We are deeply appreciative of the thoughtful feedback you have provided on our manuscript. Your expertise and detailed comments have been instrumental in enhancing our work.

As we approach the **discussion deadline**, we kindly encourage any further questions or comments you might have, particularly from reviewers cJk8 and Cavr. Our aim is to ensure that we have thoroughly addressed your concerns, and therefore, we kindly ask if you could reconsider your initial ratings.

Please rest assured that we have diligently worked to incorporate your feedback and are prepared to offer any additional clarification or information that may be required.

Once again, thank you all for your significant contributions to our manuscript's enhancement.

---

### Author Response · Authors · 2024-03-27
**Last Day of the Discussion Period**

Dear Reviewers,

I hope this message finds you well. As you know, today marks the final day to discuss any further concerns. We hope that we have addressed all of your lingering questions and are attentive to any last-minute comments.

We are also currently reviewing our document to correct any possible linguistic mistakes and ensure the clarity and accuracy of our submission.

We greatly appreciate your valuable time and insights and hope our revisions have positively influenced your initial rating.

---

### Meta-Review · Area_Chair_uj7C · 2024-04-03

**Recommendation:** Accept (Oral)
**Confidence:** 5

**Metareview:**

The authors considerably made changes to the manuscript and could convince the more critical reviewers to move from reject to accept. Three reviewers are now voting for acceptance and the topic (evaluation of synthetic images) I think is very timely and valuable for MIDL. My vote is acceptance.

---

### Decision · Program_Chairs · 2024-04-05

Accept (Oral)